# Distributed Power-law Graph Computing: Theoretical and Empirical Analysis

**Cong Xie**
Dept. of Comp. Sci. and Eng.
Shanghai Jiao Tong University
800 Dongchuan Road
Shanghai 200240, China
xcgoner1108@gmail.com

**Ling Yan**
Dept. of Comp. Sci. and Eng.
Shanghai Jiao Tong University
800 Dongchuan Road
Shanghai 200240, China
yling0718@sjtu.edu.cn

**Wu-Jun Li**
National Key Lab. for Novel Software Tech.
Dept. of Comp. Sci. and Tech.
Nanjing University
Nanjing 210023, China
liwujun@nju.edu.cn

**Zhihua Zhang**
Dept. of Comp. Sci. and Eng.
Shanghai Jiao Tong University
800 Dongchuan Road
Shanghai 200240, China
zhang-zh@cs.sjtu.edu.cn

## Abstract

With the emergence of big graphs in a variety of real applications like social networks, machine learning based on distributed graph-computing (DGC) frameworks has attracted much attention from big data machine learning community. In DGC frameworks, the graph partitioning (GP) strategy plays a key role to affect the performance, including the workload balance and communication cost. Typically, the degree distributions of natural graphs from real applications follow skewed power laws, which makes GP a challenging task. Recently, many methods have been proposed to solve the GP problem. However, the existing GP methods cannot achieve satisfactory performance for applications with power-law graphs. In this paper, we propose a novel vertex-cut method, called *degree-based hashing* (DBH), for GP. DBH makes effective use of the skewed degree distributions for GP. We theoretically prove that DBH can achieve lower communication cost than existing methods and can simultaneously guarantee good workload balance. Furthermore, empirical results on several large power-law graphs also show that DBH can outperform the state of the art.

## 1 Introduction

Recent years have witnessed the emergence of big graphs in a large variety of real applications, such as the web and social network services. Furthermore, many machine learning and data mining algorithms can also be modeled with graphs [13]. Hence, machine learning based on distributed graph-computing (DGC) frameworks has attracted much attention from big data machine learning community [13, 15, 14, 6, 11, 7]. To perform distributed (parallel) graph-computing on clusters with several machines (servers), one has to partition the whole graph across the machines in a cluster. Graph partitioning (GP) can dramatically affect the performance of DGC frameworks in terms of workload balance and communication cost. Hence, the GP strategy typically plays a key role in DGC frameworks. The ideal GP method should minimize the cross-machine communication cost, and simultaneously keep the workload in every machine approximately balanced.

Existing GP methods can be divided into two main categories: edge-cut and vertex-cut methods. Edge-cut tries to evenly assign the vertices to machines by cutting the edges. In contrast, vertex-cut tries to evenly assign the edges to machines by cutting the vertices. Figure 1 illustrates the edge-cut and vertex-cut partitioning results of an example graph. In Figure 1 (a), the edges $(A,C)$ and $(A,E)$ are cut, and the two machines store the vertex sets $\{A,B,D\}$ and $\{C,E\}$, respectively. In Figure 1 (b), the vertex $A$ is cut, and the two machines store the edge sets $\{(A,B),(A,D),(B,D)\}$ and $\{(A,C),(A,E),(C,E)\}$, respectively. In edge-cut, both machines of a cut edge should maintain a ghost (local replica) of the vertex and the edge data. In vertex-cut, all the machines associated with a cut vertex should maintain a mirror (local replica) of the vertex. The ghosts and mirrors are shown in shaded vertices in Figure 1. In edge-cut, the workload of a machine is determined by the number of vertices located in that machine, and the communication cost of the whole graph is determined by the number of edges spanning different machines. In vertex-cut, the workload of a machine is determined by the number of edges located in that machine, and the communication cost of the whole graph is determined by the number of mirrors of the vertices.

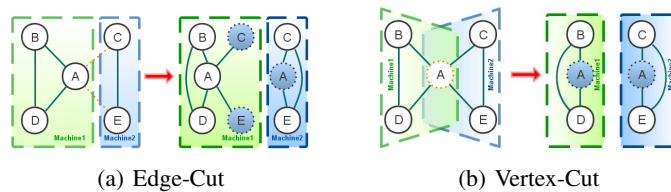

(a) Edge-Cut        (b) Vertex-Cut

Figure 1: Two strategies for graph partitioning. Shaded vertices are ghosts and mirrors, respectively.

Most traditional DGC frameworks, such as GraphLab [13] and Pregel [15], use edge-cut methods [9, 18, 19, 20] for GP. Very recently, the authors of PowerGraph [6] find that the vertex-cut methods can achieve better performance than edge-cut methods, especially for power-law graphs. Hence, vertex-cut has attracted more and more attention from DGC research community. For example, PowerGraph [6] adopts a *random* vertex-cut method and two greedy variants for GP. GraphBuilder [8] provides some heuristics, such as the *grid*-based constrained solution, to improve the random vertex-cut method.

Large natural graphs usually follow skewed degree distributions like power-law distributions, which makes GP challenging. Different vertex-cut methods can result in different performance for power-law graphs. For example, Figure 2 (a) shows a toy power-law graph with only one vertex having much higher degree than the others. Figure 2 (b) shows a partitioning strategy by cutting the vertices $\{E, F, A, C, D\}$, and Figure 2 (c) shows a partitioning strategy by cutting the vertices $\{A, E\}$. We can find that the partitioning strategy in Figure 2 (c) is better than that in Figure 2 (b) because the number of mirrors in Figure 2 (c) is smaller which means less communication cost. The intuition underlying this example is that cutting higher-degree vertices can result in fewer mirror vertices. Hence, the power-law degree distribution can be used to facilitate GP. Unfortunately, existing vertex-cut methods, including those in PowerGraph and GraphBuilder, make rarely use of the power-law degree distribution for GP. Hence, they cannot achieve satisfactory performance in natural power-law graphs. PowerLyra [4] tries to combine both edge-cut and vertex-cut together by using the power-law degree distribution. However, it is lack of theoretical guarantee.

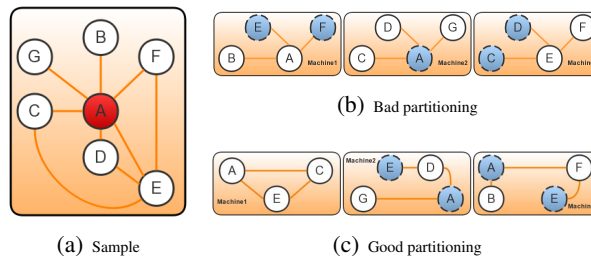

(a) Sample      (b) Bad partitioning      (c) Good partitioning

Figure 2: Partition a sample graph with vertex-cut.

In this paper, we propose a novel vertex-cut GP method, called *degree-based hashing* (DBH), for distributed power-law graph computing. The main contributions of DBH are briefly outlined as follows:

- DBH can effectively exploit the power-law degree distributions in natural graphs for vertex-cut GP.

- Theoretical bounds on the communication cost and workload balance for DBH can be derived, which show that DBH can achieve lower communication cost than existing methods and can simultaneously guarantee good workload balance.

- DBH can be implemented as an execution engine for PowerGraph [6], and hence all PowerGraph applications can be seamlessly supported by DBH.

- Empirical results on several large real graphs and synthetic graphs show that DBH can outperform the state-of-the-art methods.

## 2  Problem Formulation

Let $G = (V, E)$ denote a graph, where $V = \{v_1, v_2, \ldots, v_n\}$ is the set of vertices and $E \subseteq V \times V$ is the set of edges in $G$. Let $|V|$ denote the cardinality of the set $V$, and hence $|V| = n$. $v_i$ and $v_j$ are called neighbors if $(v_i, v_j) \in E$. The degree of $v_i$ is denoted as $d_i$, which measures the number of neighbors of $v_i$. Please note that we only need to consider the GP task for undirected graphs because the workload mainly depends on the number of edges no matter directed or undirected graphs the computation is based on. Even if the computation is based on directed graphs, we can also use the undirected counterparts of the directed graphs to get the partitioning results.

Assume we have a cluster of $p$ machines. Vertex-cut GP is to assign each edge with the two corresponding vertices to one of the $p$ machines in the cluster. The assignment of an edge is unique, while vertices may have replicas across different machines. For DGC frameworks based on vertex-cut GP, the workload (amount of computation) of a machine is roughly linear in the number of edges located in that machine, and the replicas of the vertices incur communication for synchronization. So the goal of vertex-cut GP is to *minimize* the number of replicas and simultaneously *balance* the number of edges on each machine.

Let $M(e) \in \{1, \ldots, p\}$ be the machine edge $e \in E$ is assigned to, and $A(v) \subseteq \{1, \ldots, p\}$ be the span of vertex $v$ over different machines. Hence, $|A(v)|$ is the number of replicas of $v$ among different machines. Similar to PowerGraph [6], one of the replicas of a vertex is chosen as the *master* and the others are treated as the *mirrors* of the master. We let $Master(v)$ denote the machine in which the master of $v$ is located. Hence, the goal of vertex-cut GP can be formulated as follows:

$$\min_A \frac{1}{n} \sum_{i=1}^n |A(v_i)|$$

$$s.t. \ \max_m |\{e \in E \mid M(e) = m\}| < \lambda \frac{|E|}{p} \ , \text{ and } \ \max_m |\{v \in V \mid Master(v) = m\}| < \rho \frac{n}{p},$$

where $m \in \{1, \ldots, p\}$ denotes a machine, $\lambda \geq 1$ and $\rho \geq 1$ are imbalance factors. We define $\frac{1}{n} \sum_{i=1}^n |A(v_i)|$ as *replication factor*, $\frac{p}{|E|} \max_m |\{e \in E \mid M(e) = m\}|$ as *edge-imbalance*, and $\frac{p}{n} \max_m |\{v \in V \mid Master(v) = m\}|$ as *vertex-imbalance*. To get a good balance of workload, $\lambda$ and $\rho$ should be as small as possible.

The degrees of natural graphs usually follow skewed power-law distributions [3, 1]:

$$\Pr(d) \propto d^{-\alpha},$$

where $\Pr(d)$ is the probability that a vertex has degree $d$ and the power parameter $\alpha$ is a positive constant. The lower the $\alpha$ is, the more skewed a graph will be. This power-law degree distribution makes GP challenging [6]. Although vertex-cut methods can achieve better performance than edge-cut methods for power-law graphs [6], existing vertex-cut methods, such as *random* method in PowerGraph and *grid*-based method in GraphBuilder [8], cannot make effective use of the power-law distribution to achieve satisfactory performance.

# 3 Degree-Based Hashing for GP

In this section, we propose a novel vertex-cut method, called *degree-based hashing* (DBH), to effectively exploit the power-law distribution for GP.

## 3.1 Hashing Model

We refer to a certain machine by its index $idx$, and the $idx$th machine is denoted as $P_{idx}$. We first define two kinds of hash functions: vertex-hash function $idx = vertex\_hash(v)$ which hashes vertex $v$ to the machine $P_{idx}$, and edge-hash function $idx = edge\_hash(e)$ or $idx = edge\_hash(v_i, v_j)$ which hashes edge $e = (v_i, v_j)$ to the machine $P_{idx}$.

Our *hashing model* includes two main components:

- *Master-vertex assignment*: The *master* replica of $v_i$ is uniquely assigned to one of the $p$ machines with equal probability for each machine by some *randomized* hash function $vertex\_hash(v_i)$.

- *Edge assignment*: Each edge $e = (v_i, v_j)$ is assigned to one of the $p$ machines by some hash function $edge\_hash(v_i, v_j)$.

It is easy to find that the above hashing model is a vertex-cut GP method. The *master-vertex assignment* can be easily implemented, which can also be expected to achieve a low vertex-imbalance score. On the contrary, the *edge assignment* is much more complicated. Different edge-hash functions can achieve different *replication factors* and different *edge-imbalance* scores. Please note that replication factor reflects communication cost, and edge-imbalance reflects workload-imbalance. Hence, the key of our hashing model lies in the edge-hash function $edge\_hash(v_i, v_j)$.

## 3.2 Degree-Based Hashing

From the example in Figure 2, we observe that in power-law graphs the replication factor, which is defined as the total number of replicas divided by the total number of vertices, will be smaller if we cut vertices with relatively higher degrees. Based on this intuition, we define the $edge\_hash(v_i, v_j)$ as follows:

$$edge\_hash(v_i, v_j) = \begin{cases} vertex\_hash(v_i) & \text{if } d_i < d_j, \\ vertex\_hash(v_j) & \text{otherwise.} \end{cases} \tag{1}$$

It means that we use the vertex-hash function to define the edge-hash function. Furthermore, the edge-hash function value of an edge is determined by the degrees of the two associated vertices. More specifically, the edge-hash function value of an edge is defined by the vertex-hash function value of the associated vertex with a smaller degree. Hence, our method is called *degree-based hashing* (DBH). DBH can effectively capture the intuition that cutting vertices with higher degrees will get better performance.

Our DBH method for vertex-cut GP is briefly summarized in Algorithm 1, where $[n] = \{1, \ldots, n\}$.

---
**Algorithm 1** Degree-based hashing (DBH) for vertex-cut GP
---
**Input:** The set of edges $E$; the set of vertices $V$; the number of machines $p$.
**Output:** The assignment $M(e) \in [p]$ for each edge $e$.
1: Initialization: count the degree $d_i$ for each $i \in [n]$ in parallel
2: **for all** $e = (v_i, v_j) \in E$ **do**
3:     Hash each edge in parallel:
4:     **if** $d_i < d_j$ **then**
5:         $M(e) \leftarrow vertex\_hash(v_i)$
6:     **else**
7:         $M(e) \leftarrow vertex\_hash(v_j)$
8:     **end if**
9: **end for**
---

# 4 Theoretical Analysis

In this section, we present theoretical analysis for our DBH method. For comparison, the *random* vertex-cut method (called *Random*) of PowerGraph [6] and the *grid*-based constrained solution (called *Grid*) of GraphBuilder [8] are adopted as baselines. Our analysis is based on randomization. Moreover, we assume that the graph is undirected and there are no duplicated edges in the graph. We mainly study the performance in terms of *replication factor*, *edge-imbalance* and *vertex-imbalance* defined in Section 2. Due to space limitation, we put the proofs of all theoretical results into the supplementary material.

## 4.1 Partitioning Degree-fixed Graphs

Firstly, we assume that the degree sequence $\{d_i\}_{i=1}^n$ is fixed. Then we can get the following expected replication factor produced by different methods.

*Random* assigns each edge evenly to the $p$ machines via a randomized hash function. The result can be directly got from PowerGraph [6].

**Lemma 1.** *Assume that we have a sequence of $n$ vertices $\{v_i\}_{i=1}^n$ and the corresponding degree sequence $D = \{d_i\}_{i=1}^n$. A simple randomized vertex-cut on $p$ machines has the expected replication factor:*

$$\mathbb{E}\left[\frac{1}{n}\sum_{i=1}^n |A(v_i)| \Big| D\right] = \frac{p}{n}\sum_{i=1}^n \left[1 - \left(1 - \frac{1}{p}\right)^{d_i}\right].$$

By using the *Grid* hash function, each vertex has $\sqrt{p}$ rather than $p$ candidate machines compared to *Random*. Thus we simply replace $p$ with $\sqrt{p}$ to get the following corollary.

**Corollary 1.** *By using Grid for hashing, the expected replication factor on $p$ machines is:*

$$\mathbb{E}\left[\frac{1}{n}\sum_{i=1}^n |A(v_i)| \Big| D\right] = \frac{\sqrt{p}}{n}\sum_{i=1}^n \left[1 - \left(1 - \frac{1}{\sqrt{p}}\right)^{d_i}\right].$$

Using DBH method in Section 3.2, we obtain the following result by fixing the sequence $\{h_i\}_{i=1}^n$, where $h_i$ is defined as the number of $v_i$'s adjacent edges which are hashed by the neighbors of $v_i$ according to the edge-hash function defined in (1).

**Theorem 1.** *Assume that we have a sequence of $n$ vertices $\{v_i\}_{i=1}^n$ and the corresponding degree sequence $D = \{d_i\}_{i=1}^n$. For each $v_i$, $d_i - h_i$ adjacent edges of it are hashed by $v_i$ itself. Define $H = \{h_i\}_{i=1}^n$. Our DBH method on $p$ machines has the expected replication factor:*

$$\mathbb{E}\left[\frac{1}{n}\sum_{i=1}^n |A(v_i)| \Big| H, D\right] = \frac{p}{n}\sum_{i=1}^n \left[1 - \left(1 - \frac{1}{p}\right)^{h_i+1}\right] \leq \frac{p}{n}\sum_{i=1}^n \left[1 - \left(1 - \frac{1}{p}\right)^{d_i}\right],$$

*where $h_i \leq d_i - 1$ for any $v_i$.*

This theorem says that our DBH method has smaller expected replication factor than *Random* of PowerGraph [6].

Next we turn to the analysis of the balance constraints. We still fix the degree sequence and have the following result for our DBH method.

**Theorem 2.** *Our DBH method on $p$ machines with the sequences $\{v_i\}_{i=1}^n$, $\{d_i\}_{i=1}^n$ and $\{h_i\}_{i=1}^n$ defined in Theorem 1 has the edge-imbalance:*

$$\frac{\max_m |\{e \in E \mid M(e) = m\}|}{|E|/p} = \frac{\sum_{i=1}^n \frac{h_i}{p} + \max_{j \in [p]} \sum_{v_i \in P_j} (d_i - h_i)}{2|E|/p}.$$

Although the master vertices are evenly assigned to each machine, we want to show how the randomized assignment is close to the perfect balance. This problem is well studied in the model of uniformly throwing $n$ balls into $p$ bins when $n \gg p(\ln p)^3$ [17].

**Lemma 2.** *The maximum number of master vertices for each machine is bounded as follows:*

$$\begin{cases} \Pr[MaxLoad > k_a] = o(1) & \text{if } a > 1, \\ \Pr[MaxLoad > k_a] = 1 - o(1) & \text{if } 0 < a < 1. \end{cases}$$

*Here* $MaxLoad = \max_m |\{v \in V \mid Master(v) = m\}|$, *and* $k_a = \frac{n}{p} + \sqrt{\frac{2n \ln p}{p} \left(1 - \frac{\ln \ln p}{2a \ln p}\right)}$.

## 4.2 Partitioning Power-law Graphs

Now we change the sequence of fixed degrees into a sequence of random samples generated from the power-law distribution. As a result, upper-bounds can be provided for the above three methods, which are *Random*, *Grid* and *DBH*.

**Theorem 3.** *Let the minimal degree be $d_{min}$ and each $d \in \{d_i\}_{i=1}^n$ be sampled from a power-law degree distribution with parameter $\alpha \in (2, 3)$. The expected replication factor of Random on $p$ machines can be approximately bounded by:*

$$\mathbb{E}_D \left[ \frac{p}{n} \sum_{i=1}^{n} \left(1 - \left(1 - \frac{1}{p}\right)^{d_i}\right) \right] \leq p \left[1 - \left(1 - \frac{1}{p}\right)^{\hat{\Omega}}\right],$$

*where* $\hat{\Omega} = d_{min} \times \frac{\alpha - 1}{\alpha - 2}$.

This theorem says that when the degree sequence is under power-law distribution, the upper bound of the expected replication factor increases as $\alpha$ decreases. This implies that *Random* yields a worse partitioning when the power-law graph is more skewed.

Like Corollary 1, we replace $p$ with $\sqrt{p}$ to get the similar result for *Grid*.

**Corollary 2.** *By using Grid method, the expected replication factor on $p$ machines can be approximately bounded by:*

$$\mathbb{E}_D \left[ \frac{\sqrt{p}}{n} \sum_{i=1}^{n} \left(1 - \left(1 - \frac{1}{\sqrt{p}}\right)^{d_i}\right) \right] \leq \sqrt{p} \left[1 - \left(1 - \frac{1}{\sqrt{p}}\right)^{\hat{\Omega}}\right],$$

*where* $\hat{\Omega} = d_{min} \times \frac{\alpha - 1}{\alpha - 2}$.

Note that $\sqrt{p} \left[1 - \left(1 - \frac{1}{\sqrt{p}}\right)^{\hat{\Omega}}\right] \leq p \left[1 - \left(1 - \frac{1}{p}\right)^{\hat{\Omega}}\right]$. So Corollary 2 tells us that *Grid* can reduce the replication factor but it is not motivated by the skewness of the degree distribution.

**Theorem 4.** *Assume each edge is hashed by our DBH method and $h_i \leq d_i - 1$ for any $v_i$. The expected replication factor of DBH on $p$ machines can be approximately bounded by:*

$$\mathbb{E}_{H,D} \left[ \frac{p}{n} \sum_{i=1}^{n} \left(1 - \left(1 - \frac{1}{p}\right)^{h_i + 1}\right) \right] \leq p \left[1 - \left(1 - \frac{1}{p}\right)^{\hat{\Omega}'}\right],$$

*where* $\hat{\Omega}' = d_{min} \times \frac{\alpha - 1}{\alpha - 2} - d_{min} \times \frac{\alpha - 1}{2\alpha - 3} + \frac{1}{2}$.

Note that

$$p \left[1 - \left(1 - \frac{1}{p}\right)^{\hat{\Omega}'}\right] < p \left[1 - \left(1 - \frac{1}{p}\right)^{\hat{\Omega}}\right].$$

Therefore, our DBH method can expectedly reduce the replication factor. The term $\frac{\alpha - 1}{2\alpha - 3}$ increases as $\alpha$ decreases, which means our DBH reduces more replication factor when the power-law graph is more skewed. Note that *Grid* and our DBH method actually use two different ways to reduce the replication factor. *Grid* reduces more replication factor when $p$ grows. These two approaches can be combined to obtain further improvement, which is not the focus of this paper.

Finally, we show that our DBH methd also guarantees good edge-balance (workload balance) under power-law distributions.

**Theorem 5.** *Assume each edge is hashed by the DBH method with $d_{min}$, $\{v_i\}_{i=1}^n$, $\{d_i\}_{i=1}^n$ and $\{h_i\}_{i=1}^n$ defined above. The vertices are evenly assigned. By taking the constant $2|E|/p = \mathbb{E}_D\left[\sum_{i=1}^n d_i\right] = n\mathbb{E}_D[d]/p$, there exists $\epsilon \in (0,1)$ such that the expected edge-imbalance of DBH on $p$ machines can be bounded w.h.p (with high probability). That is,*

$$\mathbb{E}_{H,D}\left[\sum_{i=1}^n \frac{h_i}{p} + \max_{j\in[p]} \sum_{v_i\in P_j}(d_i - h_i)\right] \leq (1+\epsilon)\frac{2|E|}{p}.$$

Note that any $\epsilon$ that satisfies $1/\epsilon \ll n/p$ could work for this theorem, which results in a tighter bound for large $n$. Therefore, together with Theorem 4, this theorem shows that our DBH method can reduce the replication factor and simultaneously guarantee good workload balance.

## 5 Empirical Evaluation

In this section, empirical evaluation on real and synthetic graphs is used to verify the effectiveness of our DBH method. The cluster for experiment contains 64 machines connected via 1 GB Ethernet. Each machine has 24 Intel Xeon cores and 96GB of RAM.

### 5.1 Datasets

The graph datasets used in our experiments include both synthetic and real-world power-law graphs. Each synthetic power-law graph is generated by a combination of two synthetic directed graphs. The in-degree and out-degree of the two directed graphs are sampled from the power-law degree distributions with different power parameters $\alpha$ and $\beta$, respectively. Such a collection of synthetic graphs is separated into two subsets: one subset with parameter $\alpha \geq \beta$ which is shown in Table 1(a), and the other subset with parameter $\alpha < \beta$ which is shown in Table 1(b). The real-world graphs are shown in Table 1(c). Some of the real-world graphs are the same as those in the experiment of PowerGraph. And some additional real-world graphs are from the UF Sparse Matrices Collection [5].

### Table 1: Datasets

(a) Synthetic graphs: $\alpha \geq \beta$  (b) Synthetic graphs: $\alpha < \beta$   (c) Real-world graphs

| Alias | $\alpha$ | $\beta$ | $|E|$ | | Alias | $\alpha$ | $\beta$ | $|E|$ | | Alias | Graph | $|V|$ | $|E|$ |
|-------|----------|---------|-------|-|-------|----------|---------|-------|-|-------|-------|-------|-------|
| S1 | 2.2 | 2.2 | 71,334,974 | | S10 | 2.1 | 2.2 | 88,617,300 | | Tw | Twitter [10] | 42M | 1.47B |
| S2 | 2.2 | 2.1 | 88,305,754 | | S11 | 2.0 | 2.2 | 135,998,503 | | Arab | Arabic-2005 [5] | 22M | 0.6B |
| S3 | 2.2 | 2.0 | 134,881,233 | | S12 | 2.0 | 2.1 | 145,307,486 | | Wiki | Wiki [2] | 5.7M | 130M |
| S4 | 2.2 | 1.9 | 273,569,812 | | S13 | 1.9 | 2.2 | 280,090,594 | | LJ | LiveJournal [16] | 5.4M | 79M |
| S5 | 2.1 | 2.1 | 103,838,645 | | S14 | 1.9 | 2.1 | 289,002,621 | | WG | WebGoogle [12] | 0.9M | 5.1M |
| S6 | 2.1 | 2.0 | 164,602,848 | | S15 | 1.9 | 2.0 | 327,718,498 | | | | | |
| S7 | 2.1 | 1.9 | 280,516,909 | | | | | | | | | | |
| S8 | 2.0 | 2.0 | 208,555,632 | | | | | | | | | | |
| S9 | 2.0 | 1.9 | 310,763,862 | | | | | | | | | | |

### 5.2 Baselines and Evaluation Metric

In our experiment, we adopt the *Random* of PowerGraph [6] and the *Grid* of GraphBuilder [8][1] as baselines for empirical comparison. The method *Hybrid* of PowerLyra [4] is not adopted for comparison because it combines both edge-cut and vertex-cut which is not a pure vertex-cut method.

One important metric is the replication factor, which reflects the communication cost. To test the speedup for real applications, we use the total execution time for *PageRank* which is forced to take 100 iterations. The speedup is defined as: $speedup = 100\% \times (\gamma_{Alg} - \gamma_{DBH})/\gamma_{Alg}$, where $\gamma_{Alg}$ is the execution time of PageRank with the method $Alg$. Here, $Alg$ can be *Random* or *Grid*. Because all the methods can achieve good workload balance in our experiments, we do not report it here.

### 5.3 Results

Figure 3 shows the replication factor on two subsets of synthetic graphs. We can find that our DBH method achieves much lower replication factor than *Random* and *Grid*. The replication factor of DBH is reduced by up to 80% compared to *Random* and 60% compared to *Grid*.

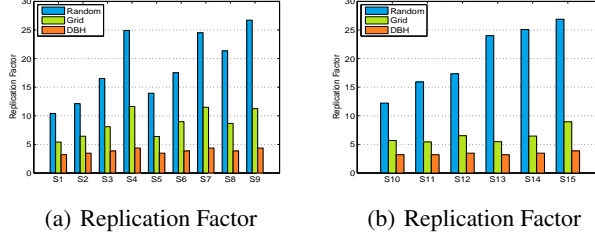

(a) Replication Factor      (b) Replication Factor

Figure 3: Experiments on two subsets of synthetic graphs. The X-axis denotes different datasets in Table 1(a) and Table 1(b). The number of machines is 48.

Figure 4 (a) shows the replication factor on the real-world graphs. We can also find that DBH achieves the best performance. Figure 4 (b) shows that the relative speedup of DBH is up to 60% over *Random* and 25% over *Grid* on the PageRank computation.

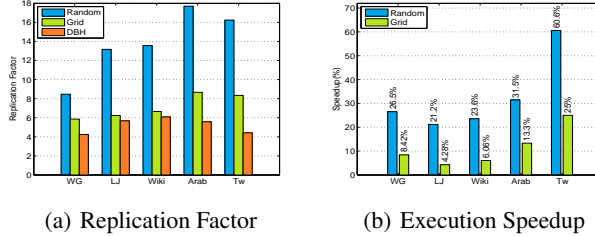

(a) Replication Factor      (b) Execution Speedup

Figure 4: Experiments on real-world graphs. The number of machines is 48.

Figure 5 shows the replication factor and execution time for PageRank on Twitter graph when the number of machines ranges from 8 to 64. We can find our DBH achieves the best performance for all cases.

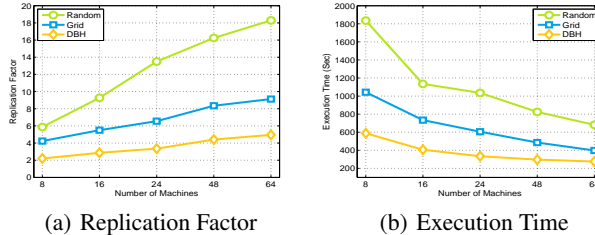

(a) Replication Factor      (b) Execution Time

Figure 5: Experiments on Twitter graph. The number of machines ranges from 8 to 64.

## 6 Conclusion

In this paper, we have proposed a new vertex-cut graph partitioning method called *degree-based hashing* (DBH) for distributed graph-computing frameworks. DBH can effectively exploit the power-law degree distributions in natural graphs to achieve promising performance. Both theoretical and empirical results show that DBH can outperform the state-of-the-art methods. In our future work, we will apply DBH to more big data machine learning tasks.

## 7 Acknowledgements

This work is supported by the NSFC (No. 61100125, No. 61472182), the 863 Program of China (No. 2012AA011003), and the Fundamental Research Funds for the Central Universities.

## Footnotes

[1]GraphLab 2.2 released in July 2013 has used PowerGraph as its engine, and the *Grid* GP method has been adopted by GraphLab 2.2 to replace the original *Random* GP method. Detailed information can be found at: http://graphlab.org/projects/index.html

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
