[Supplementary Material]

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

| Alias | $\alpha$ | $\beta$ | $|E|$ |
|---|---|---|---|
| S1 | 2.2 | 2.2 | 71,334,974 |
| S2 | 2.2 | 2.1 | 88,305,754 |
| S3 | 2.2 | 2.0 | 134,881,233 |
| S4 | 2.2 | 1.9 | 273,569,812 |
| S5 | 2.1 | 2.1 | 103,838,645 |
| S6 | 2.1 | 2.0 | 164,602,848 |
| S7 | 2.1 | 1.9 | 280,516,909 |
| S8 | 2.0 | 2.0 | 208,555,632 |
| S9 | 2.0 | 1.9 | 310,763,862 |

(b) Synthetic graphs: $\alpha < \beta$

| Alias | $\alpha$ | $\beta$ | $|E|$ |
|---|---|---|---|
| S10 | 2.1 | 2.2 | 88,617,300 |
| S11 | 2.0 | 2.2 | 135,998,503 |
| S12 | 2.0 | 2.1 | 145,307,486 |
| S13 | 1.9 | 2.2 | 280,090,594 |
| S14 | 1.9 | 2.1 | 289,002,621 |
| S15 | 1.9 | 2.0 | 327,718,498 |

(c) Real-world graphs

| Alias | Graph | $|V|$ | $|E|$ |
|---|---|---|---|
| Tw | Twitter [11] | 42M | 1.47B |
| Arab | Arabic-2005 [6] | 22M | 0.6B |
| Wiki | Wiki [2] | 5.7M | 130M |
| LJ | LiveJournal [17] | 5.4M | 79M |
| WG | WebGoogle [13] | 0.9M | 5.1M |

### 5.2 Baselines and Evaluation Metric

In our experiment, we adopt the *Random* of PowerGraph [7] and the *Grid* of GraphBuilder [9][1] as baselines for empirical comparison. The method *Hybrid* of PowerLyra [4] is not adopted for comparison because it combines both edge-cut and vertex-cut which is not a pure vertex-cut method.

One important metric is the replication factor, which reflects the communication cost. To test the speedup for real applications, we use the total execution time for *PageRank* which is forced to take 100 iterations. The speedup is defined as: $speedup = 100\% \times (\gamma_{Alg} - \gamma_{DBH})/\gamma_{Alg}$, where $\gamma_{Alg}$ is the execution time of PageRank with the method $Alg$. Here, $Alg$ can be *Random* or *Grid*. Because all the methods can achieve good workload balance in our experiments, we do not report it here.

## 5.3 Results

Figure 3 shows the replication factor on two subsets of synthetic graphs. We can find that our DBH method achieves much lower replication factor than *Random* and *Grid*. The replication factor of DBH is reduced by up to 80% compared to *Random* and 60% compared to *Grid*.

(a) Replication Factor    (b) Replication Factor

Figure 3: Experiments on two subsets of synthetic graphs. The X-axis denotes different datasets in Table 1(a) and Table 1(b). The number of machines is 48.

Figure 4 (a) shows the replication factor on the real-world graphs. We can also find that DBH achieves the best performance. Figure 4 (b) shows that the relative speedup of DBH is up to 60% over *Random* and 25% over *Grid* on the PageRank computation.

(a) Replication Factor    (b) Execution Speedup

Figure 4: Experiments on real-world graphs. The number of machines is 48.

Figure 5 shows the replication factor and execution time for PageRank on Twitter graph when the number of machines ranges from 8 to 64. We can find our DBH achieves the best performance for all cases.

(a) Replication Factor    (b) Execution Time

Figure 5: Experiments on Twitter graph. The number of machines ranges from 8 to 64.

## 6   Conclusion

In this paper, we have proposed a new vertex-cut graph partitioning method called *degree-based hashing* (DBH) for distributed graph-computing frameworks. DBH can effectively exploit the power-law degree distributions in natural graphs to achieve promising performance. Both theoretical and empirical results show that DBH can outperform the state-of-the-art methods. In our future work, we will apply DBH to more big data machine learning tasks.

## 7   Acknowledgements

This work is supported by the NSFC (No. 61100125, No. 61472182), the 863 Program of China (No. 2012AA011003), and the Fundamental Research Funds for the Central Universities.

## Footnotes

[1]GraphLab 2.2 released in July 2013 has used PowerGraph as its engine, and the *Grid* GP method has been adopted by GraphLab 2.2 to replace the original *Random* GP method. Detailed information can be found at: http://graphlab.org/projects/index.html

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

# A  Proofs

## A.1  The Proof of Theorem 1

*Proof.* Let the indicator $H_j$ denote the event that vertex $v_i$ has at least one of $h_i$ edges in the $j$th machine. Then the expectation $\mathbb{E}[H_j]$ is

$$\mathbb{E}[H_j] = 1 - \Pr(\text{none of the } h_i \text{ edges is on the machine } j)$$
$$= 1 - \left(1 - \frac{1}{p}\right)^{h_i}.$$

For some vertex $v_i$, $h_i$ adjacent edges are hashed by the neighbours of $v_i$ and $(d_i - h_i)$ adjacent edges are hashed by $v_i$ itself to the same machine. So for the $(d_i - h_i)$ edges, the number of replications of $v_i$ is simply 1 due to the assumption $h_i \leq d_i - 1$.

As for the residual $h_i$ of the adjacent edges, we have $\mathbb{E}[H_j] = 1 - \left(1 - \frac{1}{p}\right)^{h_i}$. Here $H_j$ involves the other $p - 1$ machines except for the one that already has a replication.

Putting the two parts together, we have

$$\mathbb{E}\left[|A(v_i)|\right] = 1 + \sum_{j=1}^{p-1} \mathbb{E}[H_j] = 1 + (p - 1)\left[1 - \left(1 - \frac{1}{p}\right)^{h_i}\right]$$
$$= p\left[1 - \left(1 - \frac{1}{p}\right)^{h_i + 1}\right].$$

Thus, the expected replication factor is:

$$\mathbb{E}\left[\frac{1}{n}\sum_{i=1}^{n}|A(v)|\right] = \frac{1}{n}\sum_{i=1}^{n}\left[p\left(1 - \left(1 - \frac{1}{p}\right)^{h_i + 1}\right)\right]$$
$$= \frac{p}{n}\sum_{i=1}^{n}\left[1 - \left(1 - \frac{1}{p}\right)^{h_i + 1}\right].$$

$\square$

Note that we assume $h_i \leq d_i - 1$, which means the hash function hashes at least one of the adjacent edges of $v_i$ by $v_i$ itself. Such assumption is to guarantee that there will be no "single" master vertex as which no adjacent edges are in the same partition.

By Lemma 1, we obtain

$$\frac{p}{n}\sum_{i=1}^{n}\left[1 - \left(1 - \frac{1}{p}\right)^{h_i + 1}\right] \leq \frac{p}{n}\sum_{i=1}^{n}\left[1 - \left(1 - \frac{1}{p}\right)^{d_i}\right],$$

which implies that the hash-based vertex-cut via degree-approach is at least as good as the randomized vertex-cut in the replication factor.

## A.2  The Proof of Theorem 2

*Proof.* Since we assume that the vertices are evenly hashed to all the machines, the $h_i$ adjacent edges of $v_i$ are also evenly assigned to all the machines. Subsequently, each machine has $\frac{h_i}{p}$ edges. Thus, we sum up all the vertices, obtaining $\sum_{i=1}^{n} \frac{h_i}{p}$.

For the rest $d_i - h_i$ adjacent edges of $v_i$, they are assigned to the same machine. So this part of edges incurs imbalance.

In the above procedure each edge is actually assigned twice. Thus, the final result is

$$\frac{\max\limits_{m}|\{e \in E \mid M(e) = m\}|}{|E|/p} = \frac{\sum\limits_{i=1}^{n}\frac{h_i}{p} + \max\limits_{j \in [p]}\sum\limits_{v_i \in P_j}(d_i - h_i)}{2|E|/p}.$$

$\square$

Like Theorem 1, the result in Theorem 2 also depends on the choice of hash functions. For example, if a certain hash function has always hashed a constant number of all the adjacent edges of $v_i$ by the $v_i$ itself (which means $d_i - h_i = c$ is constant and $1 \leq c < \min_i d_i$), then the expected edge-imbalance is

$$\frac{\max_m |\{e \in E \mid M(e) = m\}|}{|E|/p} = \frac{\sum_{i=1}^{n} \frac{h_i}{p} + \max_{j \in [p]} \sum_{v_i \in P_j} (d_i - h_i)}{2|E|/p} = \frac{\sum_{i=1}^{n} \frac{d_i - c}{p} + c\frac{n}{p}}{2|E|/p} = 1$$

for $\sum_{i=1}^{n} d_i = 2|E|$.

Note that there must be a tradeoff between the replication factor and edge-imbalance. To make the replication factor smaller, $h_i$ must be reduced. As a result, a smaller $h_i$ yields larger edge-imbalance according to Theorems 1 and 2.

### A.3 The Proof of Theorem 3

*Proof.* Before proving the following theorems, we introduce some properties of the power-law distribution. Note that although the degree distribution of a random graph should take integers, it is common to approximate the power-law degree distribution by continuous real numbers, which is suggested by [5]. Furthermore, although the maximum degree is $n - 1$, for convenience we assume that the maximum degree approaches infinity as $n$ is very large.

**Lemma 3.** *The power-law distribution is defined as:*
$$\Pr(d = x) = p(x) = (\alpha - 1)x_{min}^{\alpha - 1}x^{-\alpha}, \quad \text{for } x \geq x_{min}. \tag{2}$$
*The corresponding CDF (cumulative distribution function) is*
$$\Pr(d \leq x) = F(x) = 1 - \frac{x^{1-\alpha}}{x_{min}^{1-\alpha}}. \tag{3}$$

*The kth moment is*
$$\mathbb{E}\left[x^k\right] = x_{min} \times \frac{\alpha - 1}{\alpha - 1 - k}, \quad \text{for } \alpha > k + 1. \tag{4}$$

We then introduce an important inequality for our proof.

**Lemma 4.**
$$b^x \geq \theta x + c, \quad \text{for } b \in (0, 1), \quad x \geq 0, \quad \theta = b^\Omega \ln b, \quad c = b^\Omega - \Omega b^\Omega \ln b, \tag{5}$$

where $\Omega$ can be any positive value. Now we return to the proof.

Let $b = 1 - 1/p$. By Eqn. (5) we have
$$1 - \left(1 - \frac{1}{p}\right)^{d_i} \leq (1 - c) - \theta d_i.$$

Taking expectation on the degree sequence $D = \{d_i\}_{i=1}^{n}$ and by Eqn. (4), we get
$$\mathbb{E}_D\left[1 - \left(1 - \frac{1}{p}\right)^{d_i}\right] \leq \mathbb{E}_D\left[(1 - c) - \theta d_i\right]$$
$$= (1 - c) - \theta \mathbb{E}_D\left[d_i\right] = (1 - c) - \theta d_{min} \times \frac{\alpha - 1}{\alpha - 2}.$$

Since $d_i$ are i.i.d, we obtain the result:
$$\mathbb{E}_D\left[\frac{p}{n}\sum_{i=1}^{n}\left(1 - \left(1 - \frac{1}{p}\right)^{d_i}\right)\right] = \frac{p}{n}\sum_{i=1}^{n}\mathbb{E}_D\left[1 - \left(1 - \frac{1}{p}\right)^{d_i}\right] \leq p\left[1 - c - \theta d_{min} \times \frac{\alpha - 1}{\alpha - 2}\right],$$

where $c = (1 - \frac{1}{p})^\Omega - \Omega(1 - \frac{1}{p})^\Omega \ln(1 - \frac{1}{p})$ and $\theta = (1 - \frac{1}{p})^\Omega \ln(1 - \frac{1}{p})$. We take the derivative with respect to $\Omega$ to get the tightest bound. Thus when $\Omega = d_{min}(\frac{\alpha - 1}{\alpha - 2})$, the derivative is 0 and the second order derivative on this point is positive. Note that the derivative is positive if $\Omega > d_{min}(\frac{\alpha - 1}{\alpha - 2})$ and negative if $\Omega < d_{min}(\frac{\alpha - 1}{\alpha - 2})$. So $\hat{\Omega} = d_{min}(\frac{\alpha - 1}{\alpha - 2})$ is a global minimizer. Finally we can have:
$$\mathbb{E}_D\left[\frac{p}{n}\sum_{i=1}^{n}\left(1 - \left(1 - \frac{1}{p}\right)^{d_i}\right)\right] \leq p\left[1 - \left(1 - \frac{1}{p}\right)^{\hat{\Omega}}\right],$$

where $\hat{\Omega} = d_{min} \times \frac{\alpha - 1}{\alpha - 2}$. □

## A.4 The Proof of Theorem 4

*Proof.* First we view the two sequences $D = \{d_i\}_{i=1}^n$ and $H = \{h_i\}_{i=1}^n$ as constant values. By Eqn. (5) we have:

$$1 - \left(1 - \frac{1}{p}\right)^{h_i+1} \le (1-c) - \theta(h_i + 1).$$

Then we take the expectation on $\{h_i\}_{i=1}^n$ conditional on $\{d_i\}_{i=1}^n$, leading to

$$\mathbb{E}_H\left[1 - \left(1 - \frac{1}{p}\right)^{h_i+1}\Big|D\right] \le \mathbb{E}_H\left[(1-c) - \theta(h_i+1)\big|D\right]$$
$$= (1-c) - \theta\mathbb{E}_H\left[(h_i+1)\big|D\right].$$

Note that under our assumption, each of the adjacent edges is chosen to be hashed by $v_i$ with probability $\Pr(d \ge d_i) = 1 - \Pr(d \le d_i) = \frac{d_i^{1-\alpha}}{d_{min}^{1-\alpha}}$. Thus, by the assumption $h_i \le d_i - 1$, we have

$$\mathbb{E}_H\left[d_i - h_i\big|D\right] = 1 + (d_i - 1) \times \frac{d_i^{1-\alpha}}{d_{min}^{1-\alpha}},$$

which means

$$\mathbb{E}_H\left[h_i+1\big|D\right] = d_i - (d_i - 1) \times \frac{d_i^{1-\alpha}}{d_{min}^{1-\alpha}} = d_i - \frac{d_i^{2-\alpha}}{d_{min}^{1-\alpha}} + \frac{d_i^{1-\alpha}}{d_{min}^{1-\alpha}}.$$

Then we further take the expectation on $D = \{d_i\}_{i=1}^n$:

$$\mathbb{E}_D\left[d_i - \frac{d_i^{2-\alpha}}{d_{min}^{1-\alpha}} + \frac{d_i^{1-\alpha}}{d_{min}^{1-\alpha}}\right] = \int_{d_{min}}^{+\infty}\left(d_i - \frac{d_i^{2-\alpha}}{d_{min}^{1-\alpha}} + \frac{d_i^{1-\alpha}}{d_{min}^{1-\alpha}}\right)(\alpha-1)d_{min}^{\alpha-1}d^{-\alpha}\mathbf{d}d_i$$
$$= d_{min} \times \frac{\alpha-1}{\alpha-2} - d_{min} \times \frac{\alpha-1}{2\alpha-3} + \frac{1}{2}.$$

Finally, we obtain the result:

$$\mathbb{E}_{H,D}\left[\frac{p}{n}\sum_{i=1}^n\left(1 - \left(1 - \frac{1}{p}\right)^{h_i+1}\right)\right] = \frac{p}{n}\sum_{i=1}^n\mathbb{E}_{H,D}\left[1 - \left(1 - \frac{1}{p}\right)^{h_i+1}\right]$$
$$\le p\left[1 - c - \theta\left(d_{min} \times \frac{\alpha-1}{\alpha-2} - d_{min} \times \frac{\alpha-1}{2\alpha-3} + \frac{1}{2}\right)\right].$$

Here $c = \left(1 - \frac{1}{p}\right)^\Omega - \Omega\left(1 - \frac{1}{p}\right)^\Omega \ln(1 - \frac{1}{p})$ and $\theta = \left(1 - \frac{1}{p}\right)^\Omega \ln\left(1 - \frac{1}{p}\right)$. The expectation above is taken w.r.t. both $\{h_i\}_{i=1}^n$ and $\{d_i\}_{i=1}^n$.

Similar to Theorem 3, we take derivative w.r.t. $\Omega$ and get the global minimizer $\hat{\Omega}' = \left(d_{min} \times \frac{\alpha-1}{\alpha-2} - d_{min} \times \frac{\alpha-1}{2\alpha-3} + \frac{1}{2}\right)$. Thus we get the tightest bound:

$$\mathbb{E}_{H,D}\left[\frac{p}{n}\sum_{i=1}^n\left(1 - \left(1 - \frac{1}{p}\right)^{h_i+1}\right)\right] \le p\left[1 - \left(1 - \frac{1}{p}\right)^{\hat{\Omega}'}\right],$$

where $\hat{\Omega}' = \left(d_{min} \times \frac{\alpha-1}{\alpha-2} - d_{min} \times \frac{\alpha-1}{2\alpha-3} + \frac{1}{2}\right)$.

Note that our assumption suggests $d_{min} \ge 2$. And for $\alpha \in (2,3)$, $-d_{min}\frac{\alpha-1}{2\alpha-3} + \frac{1}{2} < 0$. Thus we have

$$\left(d_{min} \times \frac{\alpha-1}{\alpha-2} - d_{min} \times \frac{\alpha-1}{2\alpha-3} + \frac{1}{2}\right) < d_{min} \times \frac{\alpha-1}{\alpha-2}.$$

Thus we have:

$$p\left[1 - \left(1 - \frac{1}{p}\right)^{\hat{\Omega}'}\right] < p\left[1 - \left(1 - \frac{1}{p}\right)^{\hat{\Omega}}\right].$$

$\square$

## A.5 The Proof of Theorem 5

*Proof.* Define $x_i = (d_i - h_i)$. If $\{d_i\}_{i=1}^n$ is fixed, then $x_i \in [1, d_i]$. For specific $j \in [p]$, by Hoeffding's tail inequality, we get the following inequality conditional on the sequence $D = \{d_i\}_{i=1}^n$:

$$\Pr_H\Big\{ \sum_{v_i \in P_j} x_i \geq \mathbb{E}_H[\sum_{v_i \in P_j} x_i] + t | D \Big\} \leq \exp\Big( \frac{-2t^2}{\sum_{v_i \in P_j} (d_i - 1)^2} \Big), \quad \text{for any } t > 0.$$

Thus we obtain the inequality of the maximum value of the $p$ individual sums:

$$\Pr_H\Big\{ \max_{j \in [p]} \sum_{v_i \in P_j} x_i \leq \frac{n}{p}\mathbb{E}_H[x_i] + t | D \Big\} \geq \Big[ 1 - \exp\Big( \frac{-2t^2}{\sum_{v_i \in P_j} (d_i - 1)^2} \Big) \Big]^p.$$

Here we take $t = \epsilon \mathbb{E}_D\Big[ \sum_{v_i \in P_j} d_i \Big] = \epsilon \frac{n}{p}\mathbb{E}\big[d\big]$ where $\epsilon \in (0, 1)$. Then we have

$$\sum_{v_i \in P_j} \frac{(d_i - 1)^2}{t^2} = \sum_{v_i \in P_j} \frac{(d_i - 1)^2}{\mathbb{E}^2[d]\,\epsilon^2 n^2/p^2}.$$

Note that for $v_i \in P_j$, $\{d_i - 1\}_{i=1}^n$ is also a sequence of values under power-law distribution. When $n \to \infty$, the number of any specific value of degree $\#d_i \simeq n \Pr(d_i)$. Thus we have

$$\lim_{n \to \infty} \sum_{v_i \in P_j} \frac{(d_i - 1)^2}{n^2/p^2} \simeq \lim_{n \to \infty} \int_{d_{min}}^{n-1} \frac{(d-1)^2 n \Pr(d)}{n^2}\, \mathrm{d}d.$$

By our assumption $\alpha \in (2, 3)$ and $\Pr(d) \propto d^{-\alpha}$,

$$\lim_{n \to \infty} \int_{d_{min}}^{n-1} \frac{(d-1)^2 n \Pr(d)}{n^2}\, \mathrm{d}d = 0.$$

Thus for $n$ large enough, under our choice of $t$, $\Big( 1 - \exp\Big( -2t^2 / \sum_{v_i \in P_j} (d_i - 1)^2 \Big) \Big)^p$ is nearly 1. That is, $\max_{j \in [p]} \sum_{v_i \in P_j} x_i \leq \frac{n}{p}\mathbb{E}[x_i] + t$ w.h.p. (with high probability) when $n$ is very large. Namely,

$$\max_{j \in [p]} \sum_{v_i \in P_j} x_i \leq \frac{n}{p}\mathbb{E}_H[x_i] + t = \frac{n}{p}\mathbb{E}_H[x_i] + \epsilon\frac{n}{p}\mathbb{E}[d].$$

Now we return to the imbalance factor. By using the result above, the following is gained w.h.p:

$$\sum_{i=1}^n \frac{h_i}{p} + \max_{j \in [p]} \sum_{v_i \in P_j} (d_i - h_i) \leq \sum_{i=1}^n \frac{h_i}{p} + \frac{n}{p}\mathbb{E}_H[x_i] + \epsilon\frac{n}{p}\mathbb{E}[d_i].$$

Then we take the expectation w.r.t. $\{h_i\}_{i=1}^n$ and $\{d_i\}_{i=1}^n$:

$$\mathbb{E}_{H,D}\Big[ \sum_{i=1}^n \frac{h_i}{p} + \max_{j \in [p]} \sum_{v_i \in P_j} (d_i - h_i) \Big]$$
$$\leq \mathbb{E}_{H,D}\Big[ \sum_{i=1}^n \frac{h_i}{p} \Big] + \frac{n}{p}\mathbb{E}_{H,D}[d_i - h_i] + \epsilon\frac{n}{p}\mathbb{E}_{H,D}[d]$$
$$= (1 + \epsilon)n\mathbb{E}_{H,D}[d]/p$$
$$= (1 + \epsilon)n\mathbb{E}_D[d]/p$$

which completes the proof. $\qquad\qquad\qquad\qquad\qquad\qquad\qquad\qquad\qquad\qquad\qquad\qquad\square$