[Reviews · NeurIPS 2014]

Submitted by Assigned_Reviewer_14

1. The authors suggest a method for vertex-cut-based graph partitioning for distributed computation. Targeting good performance on naturally occurring power-law graphs, they introduce a hash-based method that works (in its simplest form) as follows:
Every edge e is uniquely assigned to a machine by selecting the incident vertex with the smallest degree and hashing that vertex's id. This intuitively leads to a balanced partitioning. The authors further provide analysis showing that their method performs at least as good as currently used graph partitioning schemes. Finally, experimental evaluation shows that for the task of PageRank, their method actually outperforms the state of the art.

Pros:
The idea is simple and clear. The solution is intuitive and apparently effective. The figures contribute to readability.

Simple analysis guarantees that the method is definitely not a bad idea and the experiments show strict improvement over the state of the art.

The authors have performed a number of experiments, however sticking to PageRank exclusively.

The suggest GP method pertains to the very timely topic of graph-computation, it is very easy to implement and it is reasonable to assume that it will have impact on the community.

Cons:
The biggest issue with the paper is organization and readability. While there are crisp ideas in the material, the presentation is lacking. Ideas do not flow well and there are some grammatical errors. Should the paper be accepted it would need a lot of editing.

The idea itself is not the big contribution; it feels like a very reasonable approach. It’s the experimental evaluation and, secondarily the analysis, that give this paper merit.
Summary: Unsurprising solution to a very timely problem. The analysis and experiments are very welcome. Paper could have impact to graph-computation community, but needs a lot of editing.

Submitted by Assigned_Reviewer_18

The paper considers the problem of partitioning graphs for
distributed machine learning. In particular, it addresses
this problem under a power-law assumption for the underlying graph.

Many commonly employed graph partitioning approaches do
not scale well to large graphs. Or, they just randomly
partition the graph (as in Pregel e.g.). That is, working
on how to partition big graphs efficiently is indeed a valid
research direction.

However, this also depends on how large the graph is. For instance
parallelized multi-level partitioning solutions such
as Pt-Scotch scaled to tens of millions of nodes. Moreover,
recent papers such as

Lu Wang, Yanghua Xiao, Bin Shao, and Haixun Wang
How to Partition a Billion-Node Graph
ICDE 2014

Charalampos E. Tsourakakis, Christos Gkantsidis, Bozidar Radunovic,
Milan Vojnovic: FENNEL: streaming graph partitioning for massive
scale graphs. WSDM 2014: 333-342

seems (to an informed outsider such as the reviewer) to
show similar performance as the presented approach (although this
is difficult to say). However, in contrast to these approaches,
the presented method is significantly simpler. This is a great benefit.

And the connection to

Florian Bourse, Marc Lelarge, and Milan Vojnovic,
Balanced Graph Edge Partition, in ACM KDD 2014

is also unclear, however, please note that KDD has not happened
yet this year.

Anyhow, all the publications show that the topic is of high
relevance. And the experiments compare to a strong baseline.
That is, the overall contribution appears to be interesting.
Summary: Touches a very important problem for big data machine learning. The major downside is that the
approach is not evaluated for machine learning problems. But partitioning is an hot topic.

Submitted by Assigned_Reviewer_42

The paper provides a graph partition strategy, for distributed computing, tailored specifically for power law graphs.

The key idea of the paper is always try to replicate node with the maximum degree, which requires less replication in skewed distribution. (this is very much expected).
The authors further provide a simple deterministic scheme to assign edges to the adjacent node with lesser degree.

In this simple framework the authors provide upper bounds on theoretical expectation of the replication factor, which turn out to be better than the grid and random approach.

In addition the authors also provide upper bounds on the expected edge imbalance.

The paper provides extensive experiments on various real datasets for computing pagerank and clearly show the advantage of the proposed simple technique over existing methods.

Summary: The overall goal is to minimize replications while balancing the edge and the vertex imbalance. The paper only has theoretical results for replications. A bound for edge imbalance is shown, which I am not sure how useful it is. It may be hard to theoretically compare all three metrics but a discussion where this balances might go worse than others would be helpful. Just showing improvement in replication is not sufficiently convincing given various measures to balance.

The strongest point of paper is extensive experiments (in supplementary) showing that this simple degree based partitioning strategy empirically outperforms exiting methods on all 3 measure.

Author Feedback
Author rebuttal: We would like to thank the reviewers for their constructive comments and suggestions. We will revise the paper based on the comments of the reviewers.

For Assigned_Reviewer_14:

Thanks a lot for the positive reviews. We will further improve the organization and readability of the paper.

For Assigned_Reviewer_18:

Thanks for your positive reviews and pointing out very recent papers related to our work.

[2.1] Graph partitioning methods can be classified into two main categories: edge-cut methods and vertex-cut methods. Both the works of Wang et al. and FENNEL of Charalampos et al. are edge-cut methods. However, our method in this paper is a vertex-cut method. Hence, our work is significantly different from the two works mentioned by the reviewer although they are related. We will add these recent papers to the references and discuss the relationship between them in the revised version.

[2.2] Indeed, the paper "Balanced Graph Edge Partition" (BGEP) has not been published yet. And we only have the access to the technical report. BGEP includes both edge-cut (vertex partition) and vertex-cut (edge partition). This is a nice work. However, BGEP is quite different from our method. More specifically, BGEP focused on the “aggregation” and presented a greedy online method. Aggregation is something like a mailbox that lumps the messages between machines to reduce the total communication cost. As stated in the paper of BGEP, the greedy online assignments do not require knowing the degree of each vertex. This implies that this greedy online method does not utilize the information of degree. But our work focuses on the degree distribution. Additionally, we feel that BGEP and our work can be combined together to further improve the performance. We will discuss the relationship between BGEP and our work in the revised version.

For Assigned_Reviewer_42:

Thanks a lot for the positive reviews.

Q: It may be hard to theoretically compare all three metrics but a discussion where this balances might go worse than others would be helpful.
A: In our paper, we assume $\epsilon$ to be $\in (0, 1)$ for the bound of edge imbalance (Theorem 5). But actually there exists a much tighter bound when $n$ (the number of vertices) is large. Note that any $1 / \epsilon << n/p$ could work in the proof of Theorem 5. Hence, when $n$ is large (e.g., 42 million in Twitter2010), we can get an $\epsilon \in (0, 0.0001)$ with high probability, which is good enough in practice. This result can also be shown in the experiments of the supplementary. Although this balance might go worse than others (e.g., Random) when $n$ is small, this will not happen in real applications because we focus on large graphs in our paper. Hence, the bound is useful to guarantee balance. Thanks! ‍